



# Hydroxyl in eclogitic garnet, orthopyroxene and oriented inclusion-bearing clinopyroxene, W Norway

Dirk Spengler[1], Monika Koch-Müller[2], Adam Włodek[1], Simon J. Cuthbert[1], and Jarosław Majka[1,3]

[1]AGH University of Krakow, Department of Mineralogy, Petrography and Geochemistry, al. A. Mickiewicza 30, 30-059 Krakow, Poland
[2]Deutsches GeoForschungsZentrum GFZ, Telegrafenberg, 14473 Potsdam, Germany
[3]Department of Earth Sciences, Uppsala University, Villavägen 16, 752-36 Uppsala, Sweden

**Correspondence:** Dirk Spengler (dirk@spengler.eu)

**Abstract.** Ten West Norwegian eclogites, whose mineral chemistry records metamorphism of up to 850 °C and 5.5 GPa, were investigated for structural hydroxyl content in nominally anhydrous minerals. Garnet shows pronounced absorption in the wavenumber ranges of 3596–3633 cm$^{-1}$, 3651–3694 cm$^{-1}$ and 3698–3735 cm$^{-1}$, and minor absorption centred at about 3560 cm$^{-1}$. Clinopyroxene with aligned inclusions of either quartz, albite or quartz + pargasite has major absorption at 3450–

3471 cm$^{-1}$ and 3521–3538 cm$^{-1}$ and minor absorption centred at 3350 cm$^{-1}$ and approximately 3625 cm$^{-1}$. The latter band is strongest in a sample with minute lamellar inclusions rich in Al, Fe and Na and was excluded from hydroxyl quantification. Orthopyroxene has large, narrow absorption peaks centred at 3415 cm$^{-1}$ and 3515 cm$^{-1}$ and smaller peaks at 3555 cm$^{-1}$, 3595 cm$^{-1}$ and 3625 cm$^{-1}$. Five orthopyroxene-bearing eclogites exhibit relatively homogeneous amounts of structural hydroxyl in garnet (13–32 µg g$^{-1}$), clinopyroxene (119–174 µg g$^{-1}$) and orthopyroxene (4–17 µg g$^{-1}$). The outer 200 µm wide

rims of the orthopyroxene grains illustrate a late hydroxyl loss compared to core values of about 30 %, which is not evident in garnet and clinopyroxene. In contrast, the other five orthopyroxene-free eclogites exhibit variable amounts of hydroxyl in garnet (8–306 µg g$^{-1}$) and clinopyroxene (58–711 µg g$^{-1}$). Apart from extreme values, the structural hydroxyl content of clinopyroxene in the eclogites studied is lower than in comparable ultra-high pressure metamorphic samples, e.g. pristine (non-metasomatised) eclogite xenoliths from the lithospheric mantle underneath the Siberian and Slave cratons (by about

200 µg g$^{-1}$) and coesite- and quartz-eclogites from the Erzgebirge and the Kokchetav massifs (by several hundred µg g$^{-1}$). The low structural hydroxyl contents, the deficiency of molecular water and the preservation of diffusion-sensitive evidence from the mineral chemistry for metamorphism well beyond the stability field of amphibole suggest that oriented inclusions of quartz + pargasite were formed isochemically during decompression. In addition, structural hydroxyl content in clinopyroxene is inversely correlated with metamorphic pressure estimates obtained from orthopyroxene of the same samples. Therefore,

structural hydroxyl in nominally anhydrous eclogite minerals can serve as an indicator of the effectiveness of retrogression.

## 1   Introduction

Nominally anhydrous minerals (NAMs) in eclogite contain crystallographically bound hydroxyl, which is an important information carrier during the evolution of the rock at high-grade metamorphism and subsequent retrogression (Gose and



Schmädicke, 2018, 2022). For example, the structural hydroxyl can indicate whether eclogite-facies hydrous minerals were
once present, whether fluid inflow occurred and whether decompression was accompanied by dehydroxylation. At the same
time, mineral chemistry, textures and inclusion microstructures are known to also carry important information, for example by
partitioning temperature- and pressure-sensitive elements between coexisting minerals (i.e. geothermometers and geobarome-
ters) or by the breakdown of unstable components from solid solutions, forming symplectic reaction textures (as in the decom-
position of jadeite) or oriented quartz inclusions (Ca-Eskola). From experimental and theoretical petrology it is known that a
change in mineral chemistry and/or the formation of a reaction texture or inclusion microstructure can each be explained by
different, sometimes contradictory, processes. However, when these features occur together in natural samples, the processes
underlying them must be consistent with the geodynamic environment of the evolution of the rocks. This reduction in ambigu-
ity is particularly beneficial for understanding the formation of oriented, notably structural hydroxyl-bearing mineral inclusions
in NAMs.

Experimental work showed that the stability field of the Ca-Eskola component ($Ca_{0.5}\square_{0.5}AlSi_2O_6$) in clinopyroxene is
almost entirely beyond that of quartz (Konzett et al., 2008a). This suggests that oriented quartz needles in natural clinopyrox-
ene from a variety of ultra-high pressure (UHP) metamorphic areas formed by isochemical exsolution during decompression
(Smith, 1984; Shatsky et al., 1985; Bakun-Czubarow, 1992; Katayama et al., 2000; Schmädicke and Müller, 2000; Dobrzhinet-
skaya et al., 2002; Song et al., 2003; Janák et al., 2004, 2013; Zhang et al., 2005). Some of the natural occurrences have such
quartz needles in close spatial association with amphibole needles, which in analogy were also proposed to be exsolved from
a former UHP clinopyroxene (Terry et al., 2000). However, the mineral minor and trace element chemistry and the inclusion
distribution provide arguments for an alternative origin of the bi-mineralic oriented inclusions. They were suggested to have
formed either by alteration and precipitation in an open system through chemical exchange with fluids or associated miner-
als (Proyer et al., 2009; Liu and Massonne, 2022) or alternatively during progressive growth of the host mineral long before
retrogression (Konzett et al., 2008b). By implication, the bi-mineralic oriented inclusions in clinopyroxene would not be an
evidence for a former Ca-Eskola component and thus formerly UHP metamorphic conditions.

The aim of this study is to determine the origin and thus the significance of lamellar amphibole occurring in close spatial
association with lamellar quartz in clinopyroxene in eclogites of the Western Gneiss Region (WGR) in Norway. For this
purpose, we quantified the structural hydroxyl content in NAMs of ten previously studied eclogites (Figure 1; the abbreviation
of mineral phases in figures, captions and the table follow the nomenclature of Warr, 2021). The eclogites have in common
that they contain clinopyroxene with aligned inclusions of either quartz, albite, or quartz + pargasite, which are thought to
have formed following UHP metamorphism, while the current mineral chemistry suggests variable metamorphic conditions
between 700–850 °C and 2.1–5.5 GPa for most of the samples (Spengler et al., 2023). In addition, we analysed the spatial
distribution of major elements in clinopyroxene in one of the ten samples. The mineral hydroxyl content is placed in context
with petrological information to evaluate possible origins for the amphibole lamellae-bearing (bi-mineralic) and amphibole
lamellae-free (mono-mineralic) oriented inclusion microstructures. We will show that the hydroxyl content of NAMs is low
and independent of the lamellar type present in the sample, but is high when hydrous minerals are present in the eclogite facies
or strong retrograde overprinting occurred.





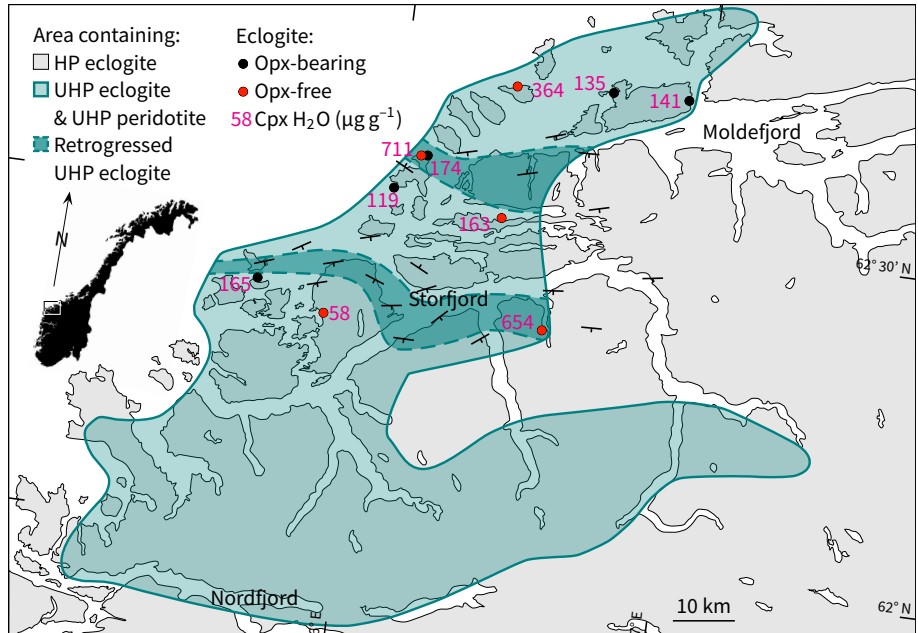

**Figure 1.** Simplified map of the WGR that shows an area with evidence for UHP metamorphism from eclogite (Smith, 1984; Wain et al., 2000; Root et al., 2005; Spencer et al., 2013; Spengler et al., 2023) and peridotite (Spengler et al., 2009, 2021) enclosed in gneiss using three approaches: index minerals (Coe, Dia), geothermobarometric estimates (exchange equilibria, net-transfer equilibria) and mineral microstructures after precursor mineral phases (polycrystalline Qz inclusions after Coe, oriented Qz inclusions after Ca-Eskola). Dots, locations of samples (this study). Labels refer to structural hydroxyl in Cpx given in Table 1. Samples with elevated structural hydroxyl in Cpx (dependent on whole-rock chemistry) are shown in dark shaded areas, whose dashed outlines were roughly extrapolated using the foliation orientation in gneiss and mylonite shown with strike and dip symbols (Young, 2018).

## 2   Geological setting and sample description

The Scandinavian Caledonides were formed during the closure of the Iapetus Ocean in the early Paleozoic and the subsequent collision of the continents Laurentia and Baltica (Gee et al., 2013). This collision caused the thrusting of nappes with peripheral, outboard and Laurentian affinities onto the Baltica plate margin, where they formed an east-verging tectono-stratigraphic succession (Gee et al., 1985). The WGR constitutes a tectonic window through this nappe pile onto the lowermost tectono-stratigraphic unit, the Lower Allochthon, which exposes high-grade metamorphic rocks with Proterozoic protolith

ages (Kullerud et al., 1986; Tucker et al., 1990). These Proterozoic Baltica basement gneisses, together with minor infolded supracrustal rocks (Krill, 1980), were reworked during the Caledonian orogeny. Radiogenic ages from high-grade quartzo-feldspathic gneiss and enclosed lenses of deformed mafic and ultramafic rocks (eclogite and pyroxenite) in the WGR suggest that maximum UHP metamorphic conditions during plate convergence in this area occurred during the final phase of the orogeny, the Scandian phase, during the Silurian and Early Devonian (Griffin and Brueckner, 1980; Carswell et al., 2003;

Tucker et al., 2004; Spengler et al., 2009; Walczak et al., 2019).



**Table 1.** FTIR absorption band average peak positions (cm$^{-1}$) and structural hydroxyl expressed as $H_2O$ equivalent (in µg g$^{-1}$) of the studied eclogite samples.

| Sample, Location | Eclogite type | Grain area | Cpx Absorption bands (1)(2)(3)(4) | Cpx Grains | Cpx $H_2O$[a] (1–3) | Cpx $H_2O$[b] (1–3) | Opx Absorption bands (1)–(6) | Opx Grains | Opx $H_2O$[a] (1–6) | Opx $H_2O$[b] (1–6) | Grt Absorption bands (1)–(7) | Grt Grains | Grt $H_2O$[a] (1–5) | Grt $H_2O$[a] (1–7) | Grt $H_2O$[b] (1–5) | Grt $H_2O$[b] (1–7) |
|---|---|---|---|---|---|---|---|---|---|---|---|---|---|---|---|---|
| 2-4A, Solholmen | Opx-bearing | rim | 3344, 3453, 3537, 3629 | 4 | 136 | 71 | 3421, 3515, 3546, 3566, 3596 | 1 | 8 | 10 | 3543, 3603, 3633, 3653, 3693, 3705, 3730 | 3 | 28 | 31 | 17 | 20 |
|  |  | core | 3344, 3454, 3538, 3626 | 5 | 146 | 78 | 3418, 3515, 3515, 3548, 3565, 3592, 3611 | 2 | 11 | 13 | 3553, 3604, 3629, 3651, 3694, 3708, 3732 | 2 | 26 | 28 | 16 | 19 |
|  |  | total |  | 6 | **141** | 75 |  | 2 | 10 | 13 |  | 5 | 27 | 29 | 17 | 20 |
| DS0326, Remøy-sunde | Opx-bearing | rim | 3453, 3527, 3621 | 4 | 221 | 124 | 3415, 3514, 3556, 3596, 3627 | 2 | 15 | 20 | 3541, 3602, 3632, 3654, 3700, 3730 | 3 | 30 | 38 | 18 | 27 |
|  |  | core | 3451, 3527, 3626 | 4 | 131 | 73 | 3415, 3514, 3556, 3596, 3628 | 2 | 18 | 23 | 3531, 3600, 3630, 3651, 3700, 3730 | 3 | 18 | 26 | 10 | 19 |
|  |  | total |  | 4 | **165** | 92 |  | 2 | 17 | 21 |  | 4 | 24 | 32 | 14 | 23 |
| DS1409, Synes | Opx-bearing | rim | 3463, 3530, 3618 | 3 | 119 | 66 | 3415, 3511, 3553, 3597, 3629 | 4 | 4 | 5 | 3558, 3597, 3630, 3657, 3675, 3700, 3731 | 4 | 7 | 14 | 4 | 12 |
|  |  | core | 3418, 3460, 3530, 3622 | 3 | 120 | 67 | 3415, 3512, 3554, 3596, 3629 | 1 | 5 | 7 | 3564, 3596, 3627, 3698, 3734 | 3 | 11 | 24 | 6 | 23 |
|  |  | total |  | 3 | **119** | 66 |  | 4 | 4 | 6 |  | 5 | 8 | 16 | 5 | 14 |
| DS2216, Langeneset | Opx-bearing | rim | 3374, 3467, 3525, 3641 | 4 | 241 | 130 | 3415, 3514, 3558, 3593, 3626 | 2 | 5 | 7 | no reliable data |  |  |  |  |  |
|  |  | core | 3558, 3457, 3521, 3633 | 1 | 40 | 23 | 3411, 3514, 3558, 3594, 3629 | 1 | 8 | 12 | no reliable data |  |  |  |  |  |
|  |  | total |  | 4 | **174** | 94 |  | 2 | 6 | 8 |  |  |  |  |  |  |
| M65, Korveneset | Opx-bearing | rim | 3450, 3526, 3636 | 2 | 162 | 88 | 3418, 3514, 3556, 3569, 3596, 3627 | 2 | 5 | 7 | 3573, 3599, 3630, 3659, 3679, 3700, 3729 | 3 | 7 | 11 | 4 | 8 |
|  |  | core | 3340, 3453, 3527, 3627 | 3 | 114 | 63 | 3417, 3515, 3556, 3591, 3616 | 3 | 8 | 10 | 3568, 3598, 3628, 3655, 3679, 3702, 3730 | 2 | 9 | 16 | 5 | 13 |
|  |  | total |  | 3 | **135** | 74 |  | 3 | 7 | 9 |  | 3 | 8 | 13 | 5 | 11 |
| DS1438, Fjørtoftvika | Opx-free ±Zo[c] | rim | 3366, 3471, 3527, 3627 | 5 | 374 | 213 |  |  |  |  | 3554, 3613, 3653, 3735 | 3 | 285 | 290 | 169 | 175 |
|  |  | core | 3338, 3470, 3527 | 2 | 358 | 209 |  |  |  |  | 3554, 3613, 3653, 3705, 3732 | 4 | 309 | 313 | 184 | 189 |
|  |  | total |  | 5 | **364** | 210 |  |  |  |  |  | 4 | 302 | 306 | 180 | 185 |
| DS1405, Riksheim | Opx-free | rim | 3379, 3468, 3523, 3629 | 5 | 697 | 373 |  |  |  |  | 3562, 3597, 3631, 3656, 3700, 3731 | 5 | 5 | 10 | 6 | 9 |
|  |  | core | 3355, 3468, 3524, 3626 | 2 | 539 | 307 |  |  |  |  | 3561, 3597, 3629, 3659, 3702, 3730 | 4 | 3 | 5 | 8 | 5 |
|  |  | total |  | 6 | **654** | 355 |  |  |  |  |  | 5 | 4 | 8 | 7 | 7 |
| DS2204, Årsetneset | Opx-free | rim | 3342, 3456, 3530, 3622 | 4 | 190 | 108 |  |  |  |  | 3568, 3596, 3629, 3655, 3681, 3699, 3729 | 3 | 5 | 7 | 3 | 6 |
|  |  | core | 3353, 3454, 3530, 3621 | 3 | 136 | 79 |  |  |  |  | 3560, 3597, 3627, 3657, 3693, 3726 | 1 | 13 | 17 | 6 | 11 |
|  |  | total |  | 4 | **163** | 93 |  |  |  |  |  | 3 | 8 | 11 | 5 | 8 |
| DS2217, Karmanns-vågen | Opx-free | rim | 3385, 3466, 3529, 3621 | 4 | 711 | 384 |  |  |  |  | 3558, 3597, 3628, 3699, 3731 | 3 | 12 | 21 | 6 | 16 |
|  |  | core | no reliable data |  |  |  |  |  |  |  | 3563, 3597, 3626, 3686, 3699, 3734 | 4 | 15 | 27 | 8 | 23 |
|  |  | total |  | 4 | **711** | 384 |  |  |  |  |  | 6 | 13 | 25 | 7 | 20 |
| UL-96-2, Ulsteinvik | Opx-free | rim | no reliable data |  |  |  |  |  |  |  | 3561, 3615, 3650, 3668, 3700, 3732 | 1 | 16 | 18 | 10 | 12 |
|  |  | core | 3470, 3622 | 1 | 58 | 31 |  |  |  |  | 3545, 3615, 3642, 3656, 3700, 3721 | 1 | 16 | 16 | 10 | 10 |
|  |  | total |  | 1 | **58** | 31 |  |  |  |  |  | 1 | 16 | 17 | 10 | 10 |

[a] Calibration of Bell et al. (1995), Cpx $H_2O$ contents in bold are those shown in Figure 1

[b] Calibration of Libowitzky and Rossman (1997)

[c] Not identified in the studied thin sections, but described in sample 1066b of Terry et al. (2000) taken from the same outcrop



Direct evidence for UHP metamorphism of gneiss is limited to a few occurrences along the coast in form of polycrystalline inclusions of quartz (inferred to be after coesite) in clinopyroxene, zoisite and clinozoisite (Wain et al., 2000), inclusions of coesite in detrital garnet (Schönig et al., 2018) and grains of diamond recovered from a crushed and dissolved sample (Do-brzhinetskaya et al., 1995). Evidence for UHP metamorphism of mafic and ultramafic rocks, exposed as isolated lenses within

gneiss, is also concentrated along the coast in terms of index mineral inclusions, i.e. coesite in eclogitic clinopyroxene, garnet and zircon (Smith, 1984; Wain, 1997; Carswell et al., 2003; Root et al., 2005), diamond in eclogitic zircon (Smith and Godard, 2013) and diamond in pyroxenitic Cr-spinel and garnet (van Roermund et al., 2002; Vrijmoed et al., 2006). However, these mafic and ultramafic rocks provide further evidence that UHP metamorphism extended spatially from the coast to the landward end of some fjords. Among them are polycrystalline inclusions of quartz in clinopyroxene and garnet from eclogite (Smith,

1984; Cuthbert et al., 2000; Walsh and Hacker, 2004) and oriented mono-mineralic inclusions of quartz in clinopyroxene (in-ferred to be after Ca-Eskola) from eclogite (Smith, 1984; Spengler et al., 2023). In addition, classical geothermobarometry of pyroxenitic and garnetitic mineral assemblages of pre-Caledonian origin shows that the former residence depth of so called Mg–Cr type ultramafites (Carswell et al., 1983) in the subcontinental lithospheric mantle for gneiss-hosted occurrences near and far from the coast was exclusively in the coesite stability field (Spengler et al., 2009, 2021). Consequently, the tectonic

transport medium (gneiss) of the ultramafites should also have been in the stability field of coesite if the model of Brueckner (1998) applies.

The ten samples analysed for hydroxyl (Table 1) come from outcrops between Storfjord and Moldefjord (nine from islands and one from the mainland, Figure 1) and were previously examined petrographically and mineralogically (Spengler et al., 2023). Five of them have the peak metamorphic mineral assemblage garnet + clinopyroxenen + orthopyroxene ± rutile ±

opaque minerals. These orthopyroxene-bearing eclogites contain clinopyroxene with bi-mineralic oriented inclusions of quartz + pargasite (Figure 2a, b). The other five samples have the peak metamorphic mineral assemblage garnet + clinopyroxene ± $SiO_2$ (coesite) ± rutile ± opaque minerals ± kyanite ± apatite. One of these orthopyroxene-free samples (DS1438) comes from an outcrop that is reported to contain minor hydrous minerals as part of the peak UHP metamorphic mineral assemblage (zoisite and phengite in sample 1066b of Terry et al., 2000; phengite in sample FJ-3C of Liu and Massonne, 2022, but which

could not be identified in the thin sections prepared. Since the outcrop size is only $5 \times 8 \, m^2$, we assume that sample DS1438 was in equilibrium with hydrous minerals during peak metamorphism. The investigated specimen has clinopyroxene-hosted bi-mineralic oriented inclusions of quartz + pargasite as decribed earlier (Terry et al., 2000). The other four orthopyroxene-free eclogites contrast with oriented inclusions of quartz, quartz + albite or albite, i.e. without pargasite (Figure 2c, d). One of these samples (DS2217) has additional, very thin parallel lamellae in clinopyroxene that have not been described previously.

Secondary minerals (biotite, amphibole and plagioclase) occur in the sample suite in varying proportions.

The orthopyroxene-bearing samples have a pyropic ternary garnet solid solution (with endmember percentages of pyrope 42–58, grossular 9–15, almandine 31–47, spessartine 1). Orthopyroxene is enstatitic (enstatite 73–86) and clinopyroxene is diopsidic to omphacitic (jadeite + aegirine 6–26). The orthopyroxene-free eclogites have a ternary garnet solid solution with large variation in the almandine content (pyrope 18–50, grossular 21–32, almandine 17–61, spessartine 0–2). Clinopyroxene is

also diopsidic to omphacitic (jadeite + aegirine 6–46) in composition.





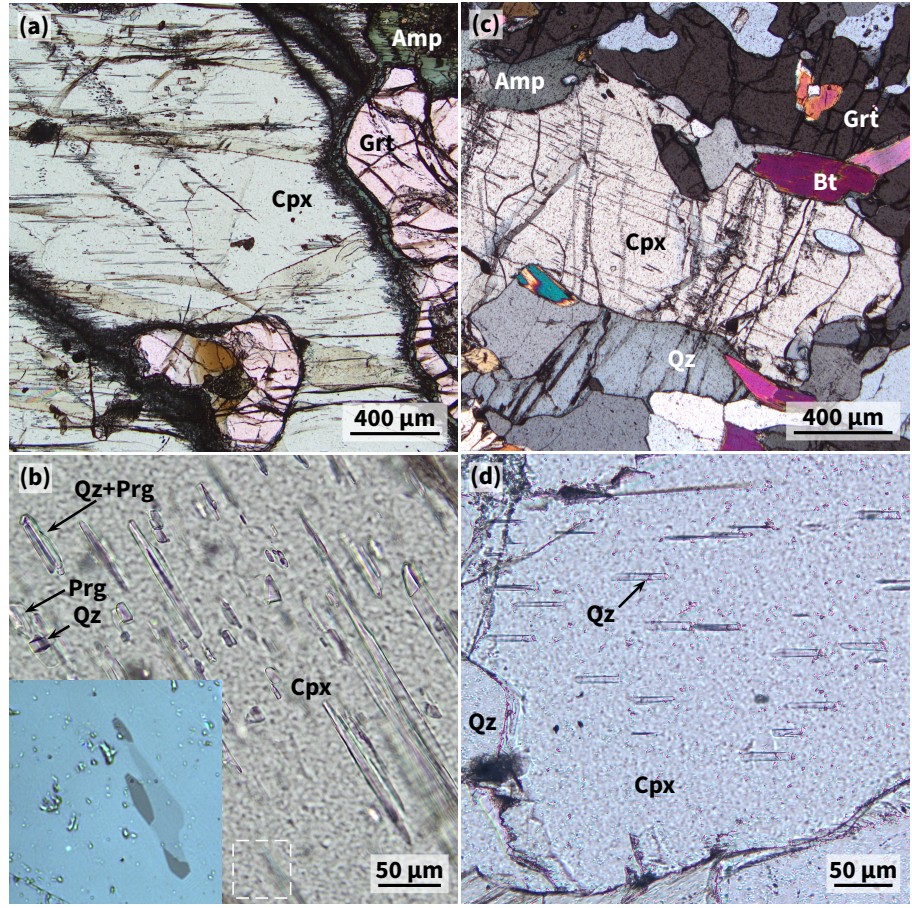

**Figure 2.** Oriented inclusion bearing Cpx in WGR eclogite. (a, b) Bi-mineralic needles (plane-polarised light, Synes eclogite DS1409). Dashed frame shows the position of the inset (reflected light). (c, d) Mono-mineralic needles (c = plane-polarised light, d = nearly cross-polarised light, Årsetneset eclogite DS2204).

## 3  Methods

Self-supporting double-polished rock slab were prepared for Fourier transform infrared spectroscopy (FTIR). The slabs were approximately $20 \times 30$ mm$^2$ in size and 180–350 µm thick. An electronic micrometre caliper was used to measure the slab thickness with a precision of 2–4 µm. These slabs were first examined by optical microscopy to locate suitable grains and grain parts that were free of cracks and inclusions or alternatively provided sufficient space in between oriented inclusions for a clear path through the minerals for analysis. Despite this approach, some of the selected sites contain oriented mono-mineralic inclusions because these could not be avoided. Unpolarised OH absorption spectra were measured using a Bruker VERTEX 80v FTIR spectrometer with an attached Hyperion2000 microscope at the GeoForschungsZentrum Potsdam. A near-infrared (NIR) light source, a CaF$_2$ beam splitter and nitrogen-cooled mercury cadmium telluride (MCT) or InSb detectors were used.






Squared apertures with a range in size from $20 \times 20\,\mu m^2$ to $100 \times 100\,\mu m^2$ (dominantly $30 \times 30\,\mu m^2$) were applied to analyse the preselected grain areas. Spectra were taken in the wavenumber range of $4000$–$2500\,cm^{-1}$ with a spectral resolution of $2\,cm^{-1}$ and averaged over 256–512 scans. Preferably, several locations inside and at the edge of each grain were analysed, unless grain geometry, the density of inclusions or fractures made this impossible.

Absorbance spectra were corrected for interference fringes, where appropriate (Neri et al., 1987), and subsequently processed
using the open-source software Fityk, version 1.3.2 (Wojdyr, 2010). Each spectrum was baseline corrected manually using a spline function and deconvoluted using a Voigtian function to determine the wavenumber of the peak $\nu$ ($cm^{-1}$) and the area (integral absorbance $A_i$) of individual absorption bands. The amount of structural hydroxyl was determined from the absorbance of the bands with peak positions in the wavenumber range of $3540$–$3340\,cm^{-1}$ for clinopyroxene, $3630$–$3410\,cm^{-1}$ for orthopyroxene and $3740$–$3530\,cm^{-1}$ / $3695$–$3530\,cm^{-1}$ for garnet. Absorption bands $>3600\,cm^{-1}$ for clinopyroxene were
attributed to the presence of nm-sized inclusions of sheet silicates (group 3 of Koch-Müller et al., 2004) and were not quantified. Where present, those centred at wavenumbers $<3500\,cm^{-1}$ for garnet were attributed to molecular water (type M of Gose and Schmädicke, 2018) and were not quantified either. The integral molar absorption coefficient $\varepsilon_i$ ($1\,mol^{-1}\,cm^{-2}$) of both Bell et al. (1995), that is mineral-specific, and Libowitzky and Rossman (1997), that is spectrum-specific and based on weighted mean wavenumbers, was used as a calibrant. The following expression of the Beer-Lambert law served for the calculation of
the structural hydroxyl content expressed as $H_2O$ equivalent $c$ (wt%):

$$c = \frac{A_{i,tot} * 1.8}{D * t * \varepsilon_i} \tag{1}$$

with $A_{i,tot}$ the total integral absorbance ($cm^{-1}$), $D$ the mineral density ($g\,cm^{-3}$) calculated from endmember proportions of the average mineral core composition (Spengler et al., 2023) and $t$ (cm) the slab thickness. Due to the use of unpolarised light, $A_{i,tot}$ (i.e. part of the numerator in equation 1) for pyroxene and garnet were approached as being equivalent to three times
$A_i$ (Konzett et al., 2008b). Since the calibration of Bell et al. (1995) requires that the $\varepsilon_i$ value (i.e. part of the denominator in equation 1) for garnet must be multiplied by three to be comparable to that of pyroxene, the factor three is cancelled out for the quantification of hydroxyl in garnet after Bell et al. (1995). Quantified hydroxyl contents were first averaged for the interior and the margin per grain and for the whole grain (by using all measurements per grain) and subsequently for each mineral per sample.

Element maps of grains of clinopyroxene from one sample were performed using a JEOL JXA-8230 electron microprobe equipped with five spectrometers for wavelength dispersive spectrometry (WDS) at the Faculty of Geology, Geophysics and Environmental Protection, AGH University of Krakow, Poland. Operating conditions were $15\,kV$ accelerating voltage and $100\,nA$ beam current. The electron beam was focused to $1\,\mu m$.

## 4 Results

### 4.1 FTIR spectra





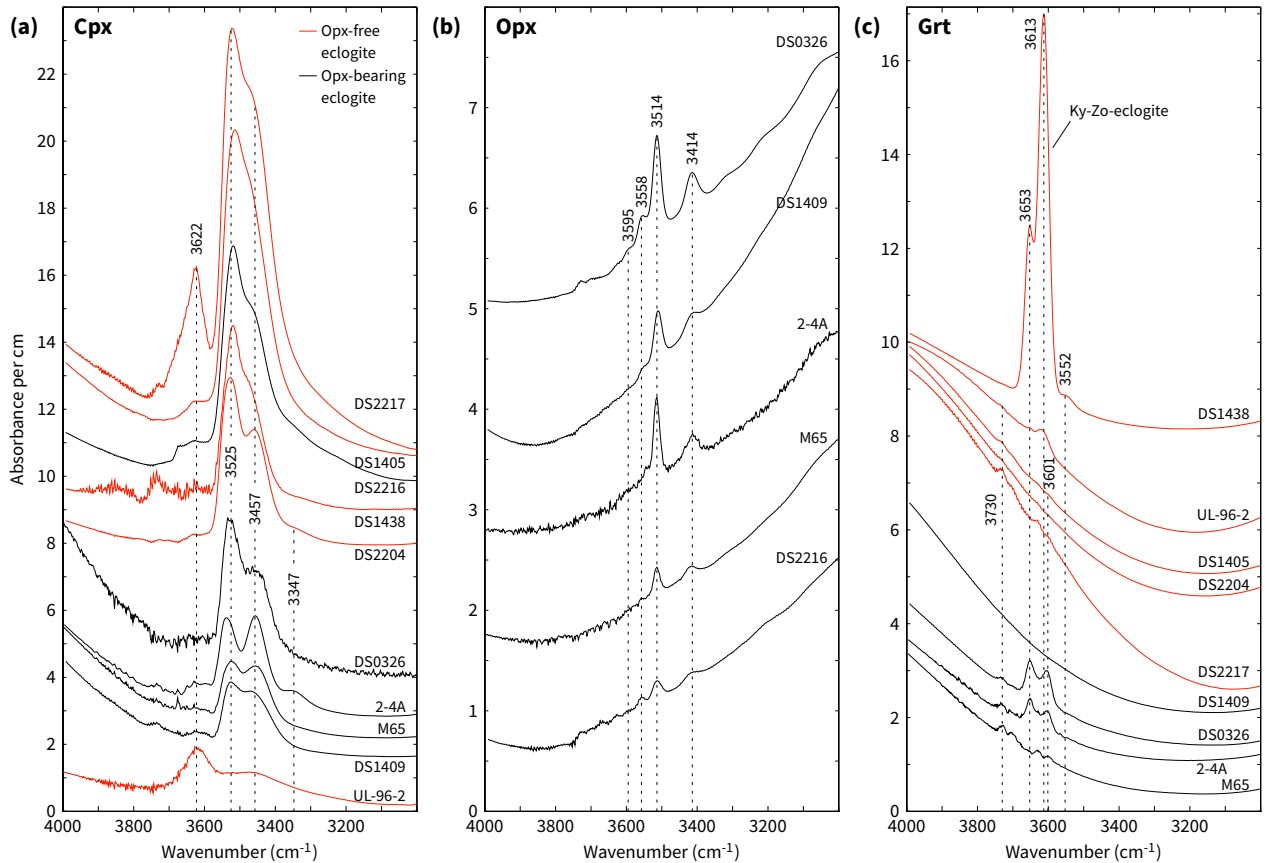

**Figure 3.** Representative unpolarised FTIR spectra (labels = samples numbers) in the O–H stetching frequency range normalised to 1 cm thickness and offset along the ordinate. Dashed lines indicate the position of selected peaks.

### 4.1.1 Clinopyroxene

Unpolarised infrared spectra of the diopsidic and omphacitic clinopyroxenes show absorption in two dominant bands at 3450–3471 cm$^{-1}$ and 3521–3538 cm$^{-1}$ for all samples except UL-96-2, where the absorption in both bands is comparably low (Figures 3a and 4a). Clinopyroxene from the latter sample exhibits an additional, pronounced absorption at higher wavenumbers in the range of 3618–3633 cm$^{-1}$, which is also observed in clinopyroxene of sample DS2217, but is weak or absent in those of the other samples. Further minor absorption at a wavenumber centred at approximately 3350 cm$^{-1}$ applies to few samples. The absorbances of the two dominant absorption bands from selected spectra show strong variation in the whole data set, between 1 and 11 for a thickness normalised to 1 cm.





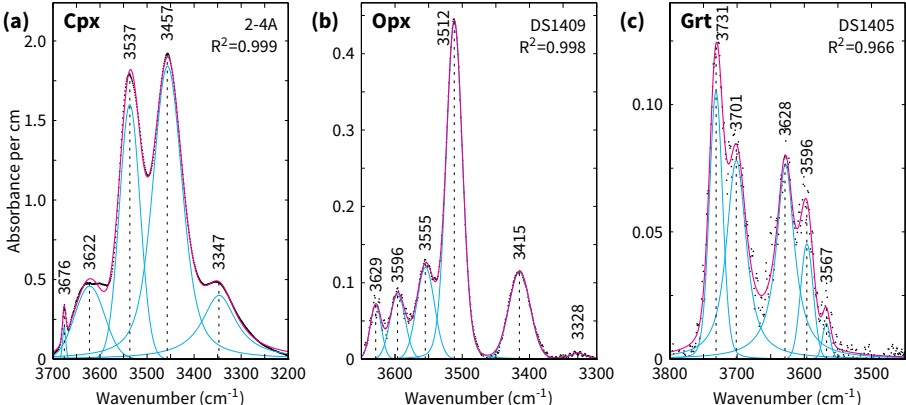

**Figure 4.** Baseline corrected and deconvoluted FTIR spectra using Voigtian functions normalised to 1 cm thickness. Dashed lines indicate the position of selected peaks.

### 4.1.2 Orthopyroxene

Orthopyroxene has two large, narrow absorption peaks centred at $3415 \, \text{cm}^{-1}$ and $3515 \, \text{cm}^{-1}$ and three to four minor peaks at higher wavenumbers centred at approximately $3555 \, \text{cm}^{-1}$, $3565 \, \text{cm}^{-1}$, $3595 \, \text{cm}^{-1}$ and $3625 \, \text{cm}^{-1}$ (Figures 3b and 4b, Table 1). Absorption at approximately $3330 \, \text{cm}^{-1}$ was rarely observed, is very small and was therefore neglected (Figure 4b). The intensities of the two dominant absorption bands vary considerably depending on the sample and are below 1 for a thickness normalised to 1 cm.

### 4.1.3 Garnet

Garnet has pronounced absorption bands with one or two peaks in each of the wavenumber ranges of $3595–3630 \, \text{cm}^{-1}$, $3650–3660 \, \text{cm}^{-1}$ and $3700–3735 \, \text{cm}^{-1}$, as well as a smaller band centred at approximately $3560 \, \text{cm}^{-1}$ (Figures 3c and 4c). All OH bands of the garnets analysed show low absorbance <1 for a thickness normalised to 1 cm. An exception is sample DS1438 from the zoisite-bearing eclogite locality (Figure 3c). If regarded separately, then garnet has rather similar variation in absorbance for orthopyroxene-bearing and orthopyroxenen-free eclogites.

### 4.2 Major element distribution in clinopyroxene of sample DS2217

Element concentration maps of clinopyroxene from sample DS2217 show that the core of the host grain between the mono-mineralic lamellae of albite is depleted in Al, Fe and Na compared to the grain periphery (Figure 5b–d). There, these elements occur spatially concentrated showing thin straight lines parallel to one of the cleavage plane directions of the host grain, which coincides with one of the albite lamellae directions and the orientation of very thin lamellar inclusions between grain rims and grain cores (Figure 5a; for higher magnification photos, the reader is referred to the previous study by Spengler et al., 2023, Figure 5e, f). These tiny inclusions between the grain rims and cores of the clinopyroxene appear to occur only in sample



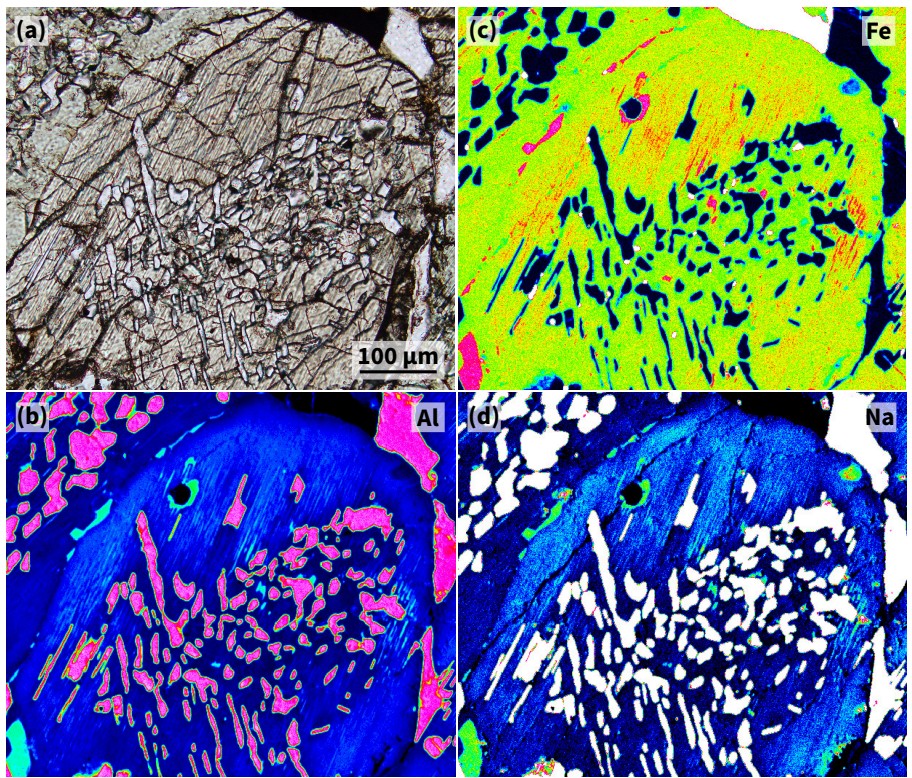

**Figure 5.** Oriented inclusion in Cpx from Karmannsvågen eclogite DS2217. (a) Photomicrograph that shows a cross-section surface of a Cpx grain (grey) with oriented lamellae and grains of Ab several µm in width (bright) in the host crystal core and lamellae of another mineral phase with sub-µm width (slightly brownish) concentrated in between core and rim of the host Cpx grain (plane-polarised light). (b–d) Compositional maps of the area shown in panel a for Al, Fe and Na (false-colour images, element concentration increases along the colour changes black–blue–green–red–white).

DS2217. They were not observed by optical methods in any other sample in the series and were not described in the previous study.

**4.3 Band assignment and hydroxyl content**

As outlined in chapter 3 we calculated the hydroxyl content of the NAMs with two independent calibrations, Bell et al. (1995) and Libowitzky and Rossman (1997). The hydroxyl contents determined with the calibration of Libowitzky and Rossman (1997) are 45±2 % and 42±4 % lower for clinopyroxene and garnet, respectively, than those of Bell et al. (1995) and 30±5 % higher for orthopyroxene. The results of both calibrations are reported in Table 1, but in order to avoid confusion we refer in 180 the following only to the values determined using the absorption coefficient of Bell et al. (1995).



### 4.3.1 Clinopyroxene

The position of four absorption bands in clinopyroxene, (1) centred at ∼3350 cm$^{-1}$, (2) 3450–3471 cm$^{-1}$, (3) 3521–3538 cm$^{-1}$ and (4) 3618–3633 cm$^{-1}$, is similar to that in synthetic and natural clinopyroxene with diopsidic, omphacitic and augitic mineral chemistry reported in earlier studies (Skogby et al., 1990; Peslier et al., 2002; Koch-Müller et al., 2004; Yang et al.,
2010; Gose and Schmädicke, 2022; Aulbach et al., 2023). There is agreement among them that the bands (2) and (3) result from the vibration of structural hydroxyl. Band (4) was shown, on the one hand, to be related to clinopyroxene from a variety of occurrences and different chemistries including that of aegirine and omphacite (Skogby et al., 1990, 2016) and can form the major absorption band in lherzolitic and wehrlitic clinopyroxenes from the Pannonian Basin (Patkó et al., 2019), but on the other, was also shown to be associated with tiny sheet silicate inclusions occurring in eclogite xenoliths of the Siberian
subcontinental lithospheric mantle (Koch-Müller et al., 2004). Given that band (4) in the current data set is dominantly minor (Figure 3a) and strongest in sample DS2217 that has optical and mineral chemical evidence for the presence of minute lamellar inclusions (Figure 5), band (4) in this study was assigned to sheet silicate inclusions and was therefore excluded from the quantification of structural hydroxyl.

The absorption band centred at approximately 3350 cm$^{-1}$ may be related to structural hydroxyl or molecular water (see
discussion in Gose and Schmädicke, 2022). This band has a low intensity in our data set (Figure 4a), does not occur in all spectra (Table 1) and was included in the quantification of hydroxyl for simplicity.

The total average structural hydroxyl content of clinopyroxene in the sample set (Table 1), calculated from bands (1–3), has the range 58–711 µg g$^{-1}$ H$_2$O equivalent using the calibration of Bell et al. (1995). If one distinguishes between the outermost 200 µm of a grain (rim) and the grain interior (core), then the ranges for rim and core are respectively 119–711 µg g$^{-1}$ and
40–539 µg g$^{-1}$.

### 4.3.2 Orthopyroxene

The six absorption bands in orthopyroxene, (1) 3411–3421 cm$^{-1}$, (2) 3511–3515 cm$^{-1}$, (3) 3546–3558 cm$^{-1}$, (4) 3565–3569 cm$^{-1}$, (5) 3591–3597 cm$^{-1}$ and (6) 3611–3629 cm$^{-1}$, were recognised to characterise a variety of igneous, metamorphic and synthetic orthopyroxenes and were ascribed to intrinsic hydroxyl (Beran and Zemann, 1986; Stalder, 2004; Patkó et al.,
2019; Tollan and Hermann, 2019). Absorption bands with wavenumbers between 3450 cm$^{-1}$ and 3630 cm$^{-1}$ were shown to correlate with trivalent Al, Cr and Fe in the crystal lattice (Stalder, 2004).

The integrated absorbances of the bands (1–6) yielded an average structural hydroxyl content in orthopyroxene in the range of 4–17 µg g$^{-1}$ H$_2$O equivalent. The range for grain rims is 4–15 µg g$^{-1}$ and that for the grain cores is slightly higher with 5–18 µg g$^{-1}$. Both sample-specific individual analyses (Figure 6b) and average values (Table 1) show that the rims systematically
have a lower hydroxyl content than the cores.



### 4.3.3 Garnet

The position of the seven absorption bands in WGR eclogite garnet, (1) 3541–3573 cm$^{-1}$, (2) 3596–3615 cm$^{-1}$, (3) 3626–3642 cm$^{-1}$, (4) 3650–3659 cm$^{-1}$, (5) 3668–3694 cm$^{-1}$, (6) 3698–3708 cm$^{-1}$ and (7) 3721–3734 cm$^{-1}$, were previously recognised in other metamorphic and synthetic garnet and were dominantly assigned to structural hydroxyl. For example, absorption

at wavenumbers within band (1) were described from Auerbach grossular (Rossman and Aines, 1991) and natural solid solutions of pyrope–almandine (Maldener et al., 2003) and almandine–grossular–pyrope (Reynes et al., 2023). Band (2) is within the range of "type II" of garnet from Erzgebirge and Fichtelgebirge eclogite, pyroxenite and peridotite and Alpe Arami peridotite (Gose and Schmädicke, 2018; Schmädicke and Gose, 2019), characterises natural and synthetic grossular–hydrogrossular garnet (Rossman and Aines, 1991), synthetic pyrope (Ackermann et al., 1983) and was assigned to the hydrogarnet substitu-

tion. Band (3) covers peaks ascribed to structural hydroxyl in synthetic pyrope (Geiger et al., 1991; Mookherjee and Karato, 2010). Band (4) is within the range of "type I" of Erzgebirge eclogite garnet (Gose and Schmädicke, 2018), denotes Roberts Victor eclogite garnet (Schmädicke et al., 2015) and synthetic pyrope with reference to the hydrogarnet substitution (Geiger et al., 1991). Wavenumbers within band (5) were documented from synthetic hydrothermally grown grossular (Geiger and Armbruster, 1997), natural grossulars in gem quality (Aines and Rossman, 1984a; Maldener et al., 2003; Reynes et al., 2018)

before and after annealing at 1000 °C (Zhang et al., 2022), pegmatitic grossular (Gadas et al., 2013) and megacryst pyropes from the Colorado Plateau, Wesselton kimberlite, Monastery kimberlite and a Cr-pyrope xenocryst from the Weltevreden kimberlite (Aines and Rossman, 1984a, b; Bell and Rossman, 1992), which support a relationship with intrinsic hydroxyl over an assignment to the presence of tiny inclusions of hydrous Mg-rich layer silicates of the serpentinite group (Geiger and Rossman, 2020). Absorption within the bands (6) and (7) is reported from almandine–grossular–pyrope solid solution garnet from

Dabieshan Bixiling eclogite (Xia et al., 2005, BXL15-1-2), Tibetan Sumdo eclogite (Gou et al., 2020, Y-4-1.7) and Cima di Gagnone eclogite (Schmädicke and Gose, 2020, CG6(1)-grt4) and were considered by the authors to be unrelated to intrinsic hydroxyl or to be indicative of secondary amphibole. On the other hand, band (6) occurs in Quebec hydrogrossular, whose intrinsic hydroxyl was constrained by mineral chemistry to occupy also other structural sites in garnet than those of the hydrogarnet substitution (Birkett and Trzcienski Jr, 1984). To address this ambiguity, the hydroxyl content of WGR eclogite

garnet was quantified with and without integral absorbances of bands (6) and (7).

The integral absorbances of the bands (1–5) of all garnets of this study revealed hydroxyl contents varying in the range of two orders of magnitude, 4–302 µg g$^{-1}$ H$_2$O equivalents. However, when excluding the sample from the zoisite-bearing eclogite (DS1438), the range shrinks to 4–27 µg g$^{-1}$. Average values for rim / core areas of grains from sample DS1438 are 285 µg g$^{-1}$ / 309 µg g$^{-1}$ and vary for grains of the other samples with 5–30 µg g$^{-1}$ / 2–26 µg g$^{-1}$, respectively.

The integral absorbances of all bands (1–7) gives hydroxyl values with a similar range of 8–306 µg g$^{-1}$, which shrinks to 8–38 µg g$^{-1}$ when DS1438 garnet is excluded. Average values for rim / core areas of DS1438 garnet are 290 µg g$^{-1}$ / 313 µg g$^{-1}$ and vary for grains of the other samples with 7–38 µg g$^{-1}$ / 6–27 µg g$^{-1}$, respectively.

## 5  Discussion



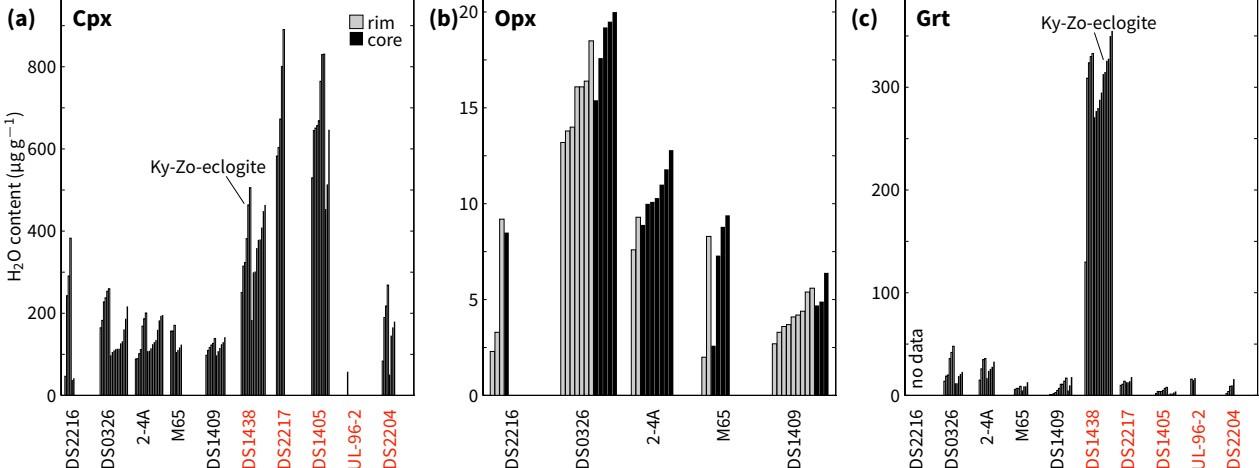

**Figure 6.** Structural hydroxyl contents expressed in $H_2O$ equivalents in the grain interior (core) and outermost 200 µm (rim) quantified using individual FTIR spectra of (a) Cpx, (b) Opx and (c) Grt using absorption bands (1–5) and the calibration of Bell et al. (1995). Numbers refer to sample numbers (black = Opx-bearing eclogite, red = Opx-free eclogite).

## 5.1 Variation of quantified hydroxyl

The approach of treating absorption bands (6) and (7) in garnet separately for the quantification of hydroxyl shows that the difference between minimum and maximum estimates, by using bands (1–5) and (1–7) respectively, for a given calibration does not exceed 17 µg g$^{-1}$ and is on average 6±4 µg g$^{-1}$ (Table 1). This rather small difference particularly affects the garnet grains with the lowest hydroxyl content, but is unlikely to be relevant for the interpretation of the data set. For reasons of simplicity, the following discussion will refer to the minimum estimates for hydroxyl in garnet based on the intergal absorbances of bands

(1–5).

Individual infrared spectra show that structural hydroxyl content in grain rims and grain cores varies for a given sample and mineral phase (Figure 6). Averages obtained separately from rims and cores also vary (Table 1). Sample-specific rim/core ratios of average hydroxyl content are below unity for orthopyroxene (Figure 7), suggesting that orthopyroxene rims experienced late hydrogen loss. Ratios for clinopyroxene cluster from unity upwards, are consistent with uptake of hydroxyl at grain rims in

some samples. Ratios for garnet cluster around unity, implying indifferent behaviour. The late hydroxyl loss in orthopyroxene should be considered when interpreting water content of this mineral from highly retrograde samples.

The hydroxyl contents determined with the calibration of Libowitzky and Rossman (1997) and that of Bell et al. (1995) differ significantly. Both calibrations base on different approaches. While the former uses wavenumber-dependent, molar absorption coefficients, the latter calibration relies on mineral-specific counterparts. In case of clinopyroxene, the absorption band at

approximately 3350 cm$^{-1}$ observed in most of the WGR samples (Table 1) is absent in the augitic clinopyroxene used in the calibration of Bell et al. (1995). Besides, the quantification after Libowitzky and Rossman (1997) has been shown to be in good agreement with $\varepsilon_i$ of Koch-Müller et al. (2007), whose structural $H_2O$ content was determined by secondary ion



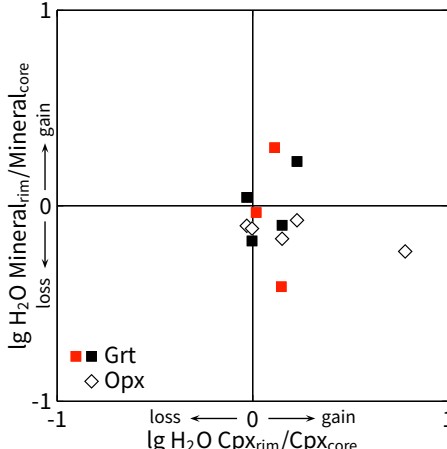

**Figure 7.** Bivariat plot that shows ratios calculated from rim and core average $H_2O$ contents in NAMs given in Table 1 (red symbols = Opx-free eclogite, other symbols = Opx-bearing eclogite) using the calibration of Bell et al. (1995) and for Grt the integrated absorbance of bands (1–5). Data values above 0 (above unity) indicate that the rims contain more $H_2O$ than the cores and vice versa.

mass spectrometry, and to be more generally applicable to clinopyroxene (Weis et al., 2018). These aspects suggest that the hydroxyl content in the analysed clinopyroxene is more likely in the range of $31–384\,\mu g\,g^{-1}$. Similar arguments could be

made for the hydroxyl contents quantified for orthopyroxene and garnet. For example, the analysed orthopyroxene shows no absorption at the wavenumbers $3060\,cm^{-1}$ and $3300\,cm^{-1}$, which were used in the calibration of Bell et al. (1995), which in turn shows no absorption at about $3625\,cm^{-1}$. All garnet absorption wavenumbers used in the calibration ($3512\,cm^{-1}$ and $3571\,cm^{-1}$) barely agree in number and position with those observed in this study. For this reason, more weight may be given to hydroxyl contents quantified using the spectrum-specific approach, and their mineral-specific counterparts may be considered

as maximum estimates. However, since the two estimates differ by less than half an order of magnitude and therefore are not expected to have any influence on the interpretation of the data set, as well as for comparison purposes, the values from Bell et al. (1995) were used in the diagrams.

## 5.2 Differences in inherited hydroxyl

There are two lines of evidence for differences in hydroxyl content inherited under conditions of peak metamorphism. One

relates to sample DS1438, whose garnet hydroxyl content is two orders of magnitude higher than that of garnet from any other sample (Figure 6c). Since this is the only sample from an outcrop studied whose peak metamorphic mineral assemblage contains hydrous minerals, zoisite and phengite (Terry et al., 2000; Liu and Massonne, 2022), the high structural hydroxyl content in its garnet is most likely related to the more hydrous conditions for the whole-rock during peak metamorphism. The hydroxyl concentration ratio between clinopyroxene and garnet in this sample is close to unity (1.2), lower than in any of the

studied samples (3.5–81.8), but similar to that of zoisite-bearing mantle eclogite (1.5–2.0, Radu et al., 2022; Figure 8).





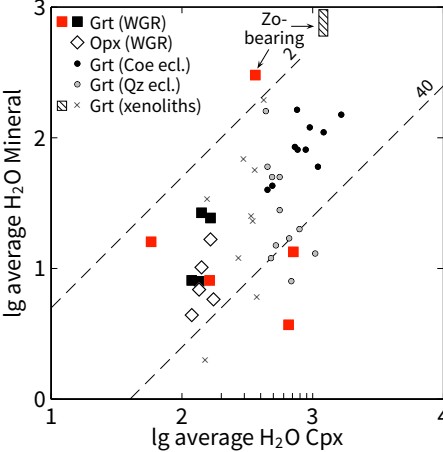

**Figure 8.** Bivariat plot that shows average $H_2O$ contents in NAMs of WGR eclogite (large symbols: red = Opx-free eclogite, others = Opx-bearing eclogite) using the calibration of Bell et al. (1995). Other data (small symbols) refer to Coe eclogite and Qz eclogite from the Erzgebirge, Fichtelgebirge and Kokchetav massif (Katayama et al., 2006; Gose and Schmädicke, 2018) and pristine (non-metasomatised) eclogite xenoliths from the Siberian and Slave cratons (Aulbach et al., 2023). Hatched field refers to a zoisite-bearing mantle exlogite xenolith (Radu et al., 2022). Dashed lines show $D_{H_2O}^{Cpx/Grt}$ ratios (labelled) that subdivide the data set. The upper (labelled 2) coincides with linearly fitted Cpx/Opx $H_2O$ concentration ratios in global peridotite xenoliths (Demouchy and Bolfan-Casanova, 2016). The lower (labelled 40) separates the samples secondarily enriched during retrogression.

The hydroxyl content of clinopyroxene provides the other indication. All those orthopyroxene-bearing eclogites, that have independent mineral-chemical evidence for equilibration at UHP metamorphic conditions (DS0326, 2-4A, M65, DS1409; Spengler et al., 2023), have cores of clinopyroxene with relatively uniform hydroxyl content (114–146 $\mu$g g$^{-1}$; Table 1). All other samples show large variation (40–539 $\mu$g g$^{-1}$). In addition, the highest hydroxyl content does not occur in clinopyroxene

of the zoisite-bearing sample (Figure 6a). This suggests that the post-peak metamorphic evolution of eclogite modified the hydroxyl content of the minerals.

### 5.3 Post-peak metamorphic hydroxyl

The Al content in orthopyroxene, which shares a mineral paragenesis with garnet, is known to sensitively record metamorphic pressures and is therefore often used as a geobarometer. The combination of such pressure estimates with the average structural

hydroxyl content of clinopyroxene, i.e. the major host for water in eclogite at the absense of hydrous minerals, shows an inverse correlation (Figure 9). This suggests that decompression led to the incorporation of hydroxyl into clinopyroxene. Since all samples are decompressed (i.e. collected from the surface), the cause is likely not the pressure change itself, but rather the mineral-chemical response to decompression (i.e. re-equilibration) that allowed clinopyroxene to incorporate additional hydroxyl. Orthopyroxene tends to show a similar inverse relationship in the metamorphic pressure range of 4.0–5.5 GPa,

which is consistent with the experimentally demonstrated dependence of the amount of structural hydroxyl on the Al content



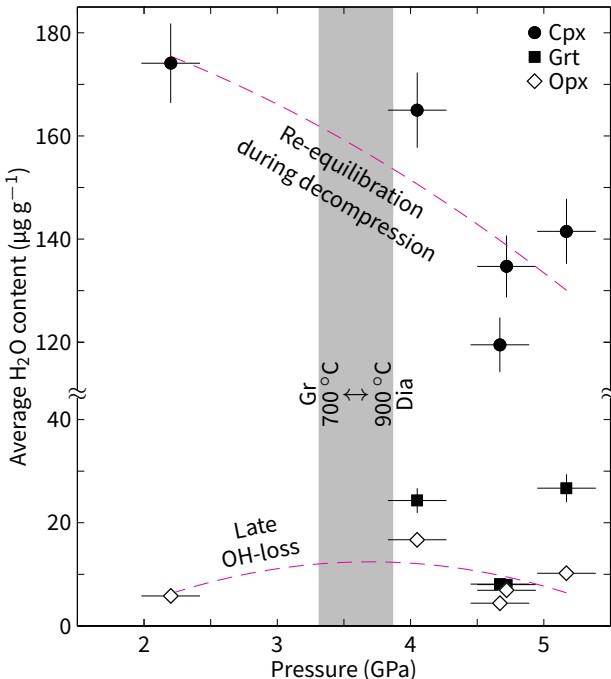

**Figure 9.** Average $H_2O$ content of NAMs in WGR Opx-bearing eclogite using the calibration of Bell et al. (1995) versus metamorphic pressure calculated from the mineral chemistry of Opx and Grt (Spengler et al., 2023). Uncertainties refer to $1\sigma$ of the calibrations used. The data for Cpx and Opx were fitted to a 2nd order polynomial (dashed lines). The Gr/Dia phase transition is shown for the temperature range 700–900 °C (grey field; Day, 2012).

(Stalder, 2004). An exception is the strongly retrogressed sample DS2216, which shows a lower than expected orthopyroxene hydroxyl content. This low content could be related to late hydrogen loss, which is particularly plausible given the fractured nature of the crystals, as this allows diffusion to efficiently affect also the crystal core regions (cf. subsection 5.1 and Figure 8c in the preceeding study).

In contrast, the orthopyroxene-free eclogites have clinoyroxene with a large variation in hydroxyl content (Figure 6a). This heterogeneity is partially related to differences in inherited hydroxyl (i.e. the presence of hydrous eclogite-facies minerals). However, the samples with the highest hydroxyl content in clinopyroxene of orthopyroxene-free eclogite (DS2217) and orthopyroxene-bearing eclogite (DS2216) are exposed only a few hundred metres apart (Figure 1). This suggests that the structural hydroxyl variation in orthopyroxene-free eclogite is in part also related to different degrees of retrogression. It follows

from the spatial proximity of chemically different samples with highest structural hydroxyl content in clinopyroxene, their lowest metamorphic pressure estimates (Figure 9) and independent textural and mineralogical evidence for strong retrogression of these samples (Spengler et al., 2023) that post-peak metamorphic retrogression could correlate with the availability of water or fluids. A major pathway for fluids are foliation planes. The rough structural extrapolation of eclogite exposure with the highest structural hydroxyl content in clinopyroxene (or degree of retrogression; samples DS1405, DS2216, DS2217) could indicate





two corridors between the coast and the hinterland in which efficient retrogression occurred (Figure 1). More samples would have to be analysed to test this hypothesis. Nevertheless, the presumed corridors are situated in between two formerly separated UHP areas (Root et al., 2005; Hacker et al., 2010) and may explain why evidence for UHP metamorphic conditions in eclogite exposed in this area was difficult to detect for a long time.

### 5.4  Lamellae formation in clinopyroxene

The source of hydroxyl structurally bound to paragasite lamellae occurring as oriented inclusions in clinoyroxene (Figure 2b) could be either external (e.g., from an infiltrating fluid) or internal (from the water content of a precursor clinopyroxene). Several arguments are against the first and in favour of the second variant. First of all, an infiltrating fluid that causes the formation of pargasite in clinopyroxene within the amphibole stability field is expected to simultaneously infiltrate associated orthopyroxene crystals. If this is true, then orthopyroxene is expected to record metamorphic conditions in the stability field

of amphibole, but not of diamond (Figure 9). Furthermore, the occurrence of bi-mineralic inclusions in clinopyroxene is independent of the structural hydroxyl content of the host mineral, which increases with retrogression and is thus a function of retrogression (Figure 9). Since the bi-mineralic inclusions occur regardless of the degree of retrogressive overprint (i.e. in peak metamorphic and retrogressed samples), their origin is unlikely to be related to the availability of external fluids.

With three exceptions (DS1438 as part of a zoisite-eclogite and DS1405 and DS2217, which contain abundant secondary

plagioclase), the clinopyroxene host has several hundred $\mu g \, g^{-1}$ fewer structural hydroxyl content than is typically found in comparable samples, e.g. coesite- and quartz-eclogite from the Erzgebirge (Gose and Schmädicke, 2018), the Kokchetav massif (Katayama et al., 2006) and pristine (non-metasomatised) eclogite xenoliths from the Siberian and Slave cratons (Aulbach et al., 2023; Figure 8). This deficite in structural water adds to the general deficite in molecular water and suggests that the mono-mineralic quartz lamellae (Figure 2c, d) formed under comparatively dry conditions, as was apparently the case for the

bi-mineralic lamellae of quartz + pargasite (Figure 2a, b). Reintegration of hydroxyl currently bound in lamellar pargasite into clinopyroxene precursor chemistry would increase its structural hydroxyl by about $200 \, \mu g \, g^{-1}$ per percent lamellar volume. The volume fraction of lamellar pargasite was not quantified. However, the low hydroxyl content of the host ($119$–$364 \, \mu g \, g^{-1}$ or less, Table 1) implies that up to several volume percent of pargasite, i.e. more than suggested by photographic images (Figure 2; Figure 3 in Spengler et al., 2023), could have theoretically exsolved from a precursor clinopyroxene, since clinopyroxene

is capable of accommodating up to more than $2000 \, \mu g \, g^{-1}$ of structural hydroxyl (Katayama and Nakashima, 2003). Fluid-mediated metasomatism, on the other hand, has been shown to form amphibole lamellae in clinopyroxene and orthopyroxene at the absence of quartz lamellae (Liptai et al., 2024). Such lamellae have not been previously reported from the samples studied, but appear to occur as parallel submicrometer-sized inclusions in clinopyroxene in sample DS2217, as indicated by a strong absorption peak at $3622 \, cm^{-1}$ (Figure 3a; Koch-Müller et al., 2004) and the spatial concentration of Na and Al typical

for amphibole and layered silicates (Figure 5). Differences in the size (thickness), mineralogy (close spatial association with quartz lamellae) and occurrence (presence in clinopyroxene but absence in orthopyroxene) of the pargasite lamellae in samples of this study differ from the microstructure of the amphibole lamellae in the samples thought to have formed by metasomatism and provide little support for the assumption that the bi-mineralic lamellae formed by a similar process.



UHP eclogites of the Kokchetav massif show that hydroxyl absorption in clinopyroxene increases with the vacancy concentration in the pyroxene structure (i.e., the Ca-Eskola component; Katayama and Nakashima, 2003). This relationship has been confirmed experimentally (Bromiley and Keppler, 2004) and suggests that samples rich in exsolved quartz lamellae may also be rich in pargasite lamellae and is consistent with a cogenetic origin of the observed bi-mineralic lamellar inclusions.

Finally, the distribution of structural hydroxyl between clinopyroxene and garnet, $D^{\mathrm{Cpx/Grt}} = c_{\mathrm{H_2O}}^{\mathrm{Cpx}} / c_{\mathrm{H_2O}}^{\mathrm{Grt}}$, is greater than 40 for the most overprinted samples (i.e. those with the highest water content in clinopyroxene: DS1405, DS2217) and significantly less than 40 for all others (Figure 8). This systematic is consistent with experimental data showing an opposite pressure dependence of the hydrogen content in both mineral phases (Aubaud et al., 2008, and references therein). By implication, the oriented mono- and bi-mineralic lamellae in the eclogitic clinopyroxene formed at high pressure and before the retrogressive overprinting that caused an increase in structural hydroxyl in clinopyroxene (Figure 9). The source of the hydrous fluid is unknown but likely related to partial melting of eclogite and embedding gneiss during advanced decompression in the quartz stability field associated with dehydration of hydrous mineral phases (e.g. zoisite and phengite; Labrousse et al., 2002; Ganzhorn et al., 2014; Cao et al., 2018; Liu and Massonne, 2022). Partial melting causes rheological weakening and thus strain partitioning. Such areas of increased deformation of gneiss have recently been shown to be more effective in retrogressive overprinting of enclosed UHP eclogite (Shulaker et al., 2024).

## 6 Conclusions

In this study on the structural hydroxyl content in clinopyroxene, orthopyroxene and garnet of WGR eclogite we come to the following conclusions:

(a) The hydroxyl content of the NAMs is low, depends on the peak metamorphic mineral assemblage and shows no systematics with the occurrence of different types of lamellar inclusions in clinopyroxene.

(b) Bi-mineralic lamellar inclusions in clinopyroxene are formed in situ by dehydration of the host mineral during decompression in the amphibole stability field rather than by an intruding hydrous component, which would presumably have obscured mineral chemical evidence of preceeding metamorphism in the stability field of diamond.

(c) An inverse correlation of structural hydroxyl in clinopyroxene containing bi-mineralic lamellae with the metamorphic pressure preserved in the associated orthopyroxene suggests a retrogressive overprint that is independent of lamellae formation. This retrogressive overprint in UHP eclogite appears to be most intense between previously recognised UHP areas.

*Author contributions.* D.S. developed the concept, measured the infrared absorption, processed the raw data, created graphics and wrote the first manuscript. M.K-M. enabled the infrared absorption measurements and ensured the quality of the data obtained. A.W. carried out the



electron microprobe work. D.S. and S.J.C. provided the samples. J.M. supported the project and the manuscript. All authors contributed to discussions and the final version of the manuscript.

*Competing interests.* The authors have no competing interests.

*Acknowledgements.* The research leading to these results has received funding from the Norwegian Financial Mechanism 2014–2021 under Project Contract No. 2020/37/K/ST10/02784 granted to D.S.



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
