# Peer review of "Hydroxyl in eclogitic garnet, orthopyroxene and oriented inclusion-bearing clinopyroxene, W Norway"

_EGUsphere, 2024_

## Author Response (AR1)

Dr. Dirk Spengler
Faculty of Geology, Geophysics and Environmental Protection
AGH University of Science and Technology
al. Mickiewicza 30
30-059 Kraków
POLAND

Prof. Dr. Johan Lissenberg
School of Earth and Environmental Sciences
Cardiff University
Park Place
Cardiff, CF10 3AT
UK

Berlin, 5th January 2025

**COVER LETTER: egusphere-2024-2734 (Spengler et al.) – re-submission**

Dear Professor Lissenberg,

Enclosed please find a manuscript entitled "Hydroxyl in eclogitic garnet, orthopyroxene and oriented inclusion-bearing clinopyroxene, W Norway", which is being re-submitted for possible publication in *Solid Earth*. My co-authors have approved the manuscript and agree to its re-submission.

This manuscript is a revised version of manuscript number egusphere-2024-2734 and is not being considered elsewhere for publication.

To strengthen the manuscript along the lines suggested by the two referees, we improved the clarity of descriptions, added requested detail and discussed, where appropriate, specific concerns risen by the experts. Supplementary Information has now been added to the manuscript to address all their concerns.

Below we comment to specific criticism or comments and explain, where necessary, how we have incorporated our responses.

Sincerely,

Dr. Dirk Spengler

**Statistics**

– Word count (abstract / main text + figure captions): 297 / 7204
– Reference count: 96
– Figure / Table / Supplementary Information count: 9 / 1 / 8
– Submitted parts: 1 manuscript file with inserted figures and tables and an attached supplement (pdf)
        1 manuscript file with text modifications highlighted in blue

**Point-by-point reply** (in black) **to comments from the referees** (in blue)

Referee no. 1

(1) There is an observed anticorrelation between hydrogen content and pressure for clinopyroxene, interpreted as reflecting retrogression conditions (during exhumation?).
The referee has correctly summarised one of the most important results of this study. We have noticed the question mark. To improve our explanation, we have added two explanatory sentences at the end of the caption of Figure 9. In addition, Figure 9 now also includes average hydroxyl data from clinopyroxene rims (Table 1) to illustrate that the anticorrelation is more pronounced when only the rims of clinopyroxene are considered. With this update of figure and caption, we hope to provide more clarity that the observed anticorrelation between metamorphic pressure and both the average hydroxyl content and the intensity of the intracrystalline hydroxyl gradient (now also shown in Supplementary Figure 4) reflects an evolutionary stage after (i.e. postdate) the formation of the oriented inclusion microstructure of quartz + pargasite (that occurs everywhere in the crystal and was inevitably formed by decompression during the early exhumation from UHP to HP).

(2) It would be insightful to compare the measured hydrogen content in this sample [DS1438] to experimental data at saturation for the given temperature and varying pressure. An agreement under UHP conditions might then suggest local H2O saturation, at least for this sample.
Unfortunately, we are not aware of any experimental study on the influence of water saturation on the hydroxyl content of nominally anhydrous minerals during UHP metamorphism. In the absence of a suggestion of such a study by the referee, we had to ignore this point.

(3) The manuscript places considerable emphasis on the presence or absence of orthopyroxene in the eclogites. However, the influence of bulk chemistry on hydrogen distribution at peak conditions receives limited exploration in the discussion. Could the chemical composition of pyroxene in differing bulk compositions partially govern hydrogen distribution?
As the hydroxyl content of orthopyroxene is lower than that of garnet and clinopyroxene, the presence or absence of orthopyroxene in the peak metamorphic mineral assemblage should have little effect on the hydroxyl distribution. To investigate the referee's point in more detail, we have plotted mineral chemical data against the hydroxyl content of clinopyroxene in Supplementary Figure 5. The bivarate diagrams show no clear correlation, with the exception of the jadeite component, which decreases with increasing hydroxyl content. This anticorrelation is consistent with an uptake of hydroxyl during the retrogression of the sample, i.e. during the partial re-equilibration of the mineral chemistry to the decompressed conditions. We have expanded the discussion accordingly by adding descriptive sentences to subsections 5.2 and 5.3.

(4) Please consider addressing the potential impact of anisotropy on the observed hydrogen content across samples with varying numbers of measured grains. For the same reason, core/rim measurements may be compromised if performed on differently oriented grains.
We thank the referee for this comment, which has clearly led to an improvement of the manuscript. In Figure 9, we have now replaced the uncertainty for hydroxyl resulting from the calibration used with the uncertainty resulting cumulatively from all analytical steps involved. The method section now contains a description of how the total uncertainty of the analytical precision was estimated. In addition, the new Supplementary Figures 3 and 4 demonstrate that single crystal intracrystalline differences between core and rim measurements lie outside the analytical uncertainty. We have added a corresponding note to subsection 5.1.

(5) Please consider adding a table (e.g., Table 1) detailing geothermobarometric constraints (presumably from Spengler et al., 2023) for samples with FTIR data, to facilitate the interpretation of Figure 9.

We have followed the referee's suggestion and have included the geothermobarometric data in Table 1.

Referee no. 2

(6) Concerning the formation of amphibole lamellae in clinopyroxene, the authors conclude that the process was driven by decompression and the consumption of internal H2O. By contrast, Konzett et al. (2008) came to the opposite conclusion, based on the H2O and trace element contents (Li, K) in eclogitic cpx from the Eastern Alps. Unfortunately, no data on mineral composition are given in the present contribution. If trace element data are available it is recommended to include such elements, especially those being indicative of fluid transfer.

The mineral trace elements were not analysed in the samples examined and are unfortunately not available. However, the previous study of Spengler et al. (2023) contains EMPA data showing that the $K_2O$ content of clinopyroxene does not exceed 0.02 wt% (analysed in spot mode, Table 4) and that of re-integrated clinopyroxene with quartz and amphibole inclusions has a range of 0.01–0.03 wt% (analysed with a broadened beam, Table 5). In addition, we had also analysed in sample DS1438, but not reported, the composition of two types of amphibole that occur as inclusions in clinopyroxene together with quartz and as a retrograde phase in the matrix between clinopyroxene and garnet. The former amphibole type contains 0.01–0.03 wt% $K_2O$ and the latter 0.07–0.17 wt%. These data suggest that the oriented inclusion-bearing clinopyroxene, the amphibole inclusions and the re-integrated clinopyroxene all contain K as a trace element component with quantities lower than those reported from the Alpine samples CM31/03 and SKP31 (Konzett et al., 2008). To address the referee's concerns, we show both textural types of amphibole in Supplementary Figure 1 and their average mineral chemistry in Supplementary Table 1 and refer to this information in the sample description (section 2) and discussion (subsection 5.4).

(7) Furthermore, the authors report that H2O in cpx is inversely correlated with pressure which they attribute to H2O incorporation during decompression. While this may be the case, there is a problem that needs to be considered. True, the experimental study by Bromiliy & Keppler (2004; cited by the authors) has shown that H2O in jadeite decreases with pressure (from 2 to 10 GPa) so that the mineral should be able to take up more H2O during decompression (compared to its content at Pmax). However, experiments with jd-di (± Ca-Eskola) solid solutions, which incorporate more H2O than pure jadeite, clearly demonstrated that clinopyroxene composition has a much greater effect on H2O content compared to P and T (Bromiley & Keppler, 2004). This is highlighted by the contrasting findings from natural cpx: H2O decrease with P (Koch-Müller et al. 2004), H2O increase (Katayama & Nakashima, 2003)), no depenence at all (Gose & Schmädicke, 2021).

The cited study of Bromiley and Keppler (2004) makes two major conclusions, i.e. the $H_2O$ storage capacity in jadeite increases with decreasing pressure and a Ca-Eskola bearing jadeite–diopside solid solution dramatically increases the $H_2O$ storage capacity in clinopyroxene. The initial decompression of WGR eclogite (from the stability fields of coesite and Ca-Eskola to that of quartz) is expected to decrease the storage capacity for $H_2O$ in clinopyroxene, from which follows that the release of quartz may be associated with that of $H_2O$. This is consistent with the joint exsolution of quartz + amphibole in WGR clinopyroxene, before further decompression caused an increase in the now Ca-Eskola-free clinopyroxene (as shown in Figure 9). The finding from Bromiley and Keppler (2004) is consistent with the other study of Katayama and Nakashima (2003) showing that the hydroxyl absorbance in clinopyroxene correlates positively (clearly increases) with the Ca-Eskola component. Conversely, this

means that the decomposition of Ca-Eskola (by decompression) is accompanied by a release of $H_2O$ (provided there was enough $H_2O$ stored in clinopyroxene prior to decompression). The study of Koch-Müller et al. (2004) shows that eclogite xenoliths from different depth levels of the Siberian lithospheric mantle differ in $H_2O$ content. Samples from the deepest levels contain less $H_2O$ in clinopyroxene than those from shallower levels. However, these measurements illustrate a systematic of the regional subcontinental lithospheric mantle (SCLM), which does not necessarily reflect a systematic for the $H_2O$ storage capacity in clinopyroxene, because the samples studied do neither share their evolution nor environment (the deep and shallow SCLM levels may chemically differ). Nevertheless, high $H_2O$ contents in clinopyroxene from the shallow Siberian SCLM are consistent with our finding that $H_2O$ increases in WGR clinopyroxene during decompression. The study of Gose and Schmädicke (2022) on eclogite from the Erzgebirge shows indeed no major difference in $H_2O$ content between HP and UHP samples, but the difference in pressure of their samples account also only to about 10 kbar and clusters at the quartz/coesite phase transition. This is very different from the WGR samples that document >30 kbar decompression, from the stability field of diamond to that of quartz. This difference in pressure seems to be large enough to show (i) the release of $H_2O$ (by formation of oriented quartz + amphibole during Ca-Eskola breakdown) and (ii) the subsequent uptake of $H_2O$ (as structural hydroxyl) in clinopyroxene during further retrogression along the exhumation path. In summary, we cannot recognise the criticism raised by the referee when looking at the details of the studies mentioned.

(8) Thus, at least the major element composition has to be included in order to justity the conclusions. This also applies to garnet. The authors report exceptionally high H2O of some 300 ppm in one sample (compared to <27 ppm in all other samples). This could simply be due to more Ca in the exceptional sample, as Ca has long been known to strongly enhance H2O (see papers by Rossman).

As requested by the referee, we have added selected information on the major element chemistry of pyroxene and garnet (Supplementary Table 1). Bivariate plots (Supplementary Figure 5) show that (i) the grossular component is similar within garnet of each of the two subgroups (orthopyroxene-bearing and orthopyroxene-free eclogite) and (ii) there is no clear correlation between the selected major element chemistry and the hydroxyl content in clinopyroxene, except for the jadeite component. This information is briefly addressed in the discussion (subsections 5.2 and 5.3).

(9) lines 10-14: The authors state that the structural hydroxyl content of clinopyroxene (some 60-700 ppm) is lower compared the literature reports for 'pristine eclogite xenoliths'. However, there are a number of such pristine samples with equally low contents (e.g., Bizimis & Peslier, 2015: 260-576 ppm; Huang et al. 2014: most samples have well below 1000 ppm).

We have expanded the literature data on eclogite xenoliths from the SCLM presented in Figure 8 to include both pristine and metasomatised xenoliths and a variety of cratonic areas to allow the reader to make an independent assessment. The diagram shows that clinopyroxene from WGR eclogite has $<200\,\mu g\,g^{-1}$ hydroxyl (unless it contains hydrous UHP minerals or was retrograded at HP), whereas clinopyroxene from comparable SCLM eclogite has almost only $>200\,\mu g\,g^{-1}$, which is also true for eclogite and pyroxenite from the cited studies. The wording in the abstract has been modified slightly accordingly.

(10) line 18-20: The authors state: 'structural hydroxyl content in clinopyroxene is inversely correlated with metamorphic pressure estimates obtained from orthopyroxene of the same samples. Therefore, structural hydroxyl ... can serve as an indicator ... of retrogression.' Clinopyroxene composition needs to be considered (see above).

This comment is similar to that of (3), which we have addressed in more detail.

(11) line 29: Decomposition of jadeite cannot result in symplectite - it just produces albite.

The referee rightly stresses that the decomposition of (pure) jadeite cannot lead to the formation of symplectite. However, what we meant, and what has been made clearer in the revised version (Introduction section), is that in practice the isochemical decomposition of the jadeite component in omphacite (which changes the composition of the pyroxene from omphacitic to diopsidic) often occurs in the form of symplectites consisting of diopsidic clinopyroxene and plagioclase (Anderson and Moecher, 2007). This is a fairly typical mineral texture for eclogite exhumed by a tectonic process as opposed to a kimberlite eruption.

(12) line 95-96: 'Since the outcrop size is only 5 × 8 m2, we assume that sample DS1438 was in equilibrium with hydrous minerals during peak metamorphism.' What has the outcrop size to do with the assemblage? This statement requires explanation.

The sentence preceding the quotation refers to "... hydrous minerals as part of the peak UHP metamorphic mineral assemblage ... could not be identified in the thin sections prepared." Therefore, we assume that the attentive reader understands without further words that due to the small outcrop size, chemical equilibration of the sample studied (DS1438) with hydrous peak metamorphic minerals can reasonably be assumed.

(13) Here, the authos list the secondary minerals (bio, amph, plag). If plag is present in decompressed eclogite, it should coexist with Na-poor cpx, unless it is heavily retrogressed in the amphibolite facies in which case all cpx (primary and secondary) is replaced by amph. However, this cannot be the case, as the authors report hydroxyl from cpx.

The referee is correct in the conclusion that clinopyroxene in sample DS2217 should be Na-poor. Table 4 of Spengler et al. (2023) shows that this clinopyroxene contains only 6–7 mol% Na-pyroxene. Therefore, we cannot recognise any critical point.

(14) pages 6-7: It is important to give the number of grains from which IR spectra were collected to determine the H2O content for each mineral in a sample. What is the uncertainty of H2O contents in anisotropic minerals and garnet?
In case that H2O is obtained from anisotropic minerals by unpolarised radiation, at least ten grains with different orientations should be analysed. Are the pyroxenes randomly oriented? If not, the approach gives unreliable data. This information has to be added.

The referee raises an important point, which is similar to that of (4), to which we responded. In addition, the number of grains from which IR spectra were collected was already given for each sample, mineral phase and grain domain in Table 1, which was probably overlooked by the referee.

(15) line 168: The authors mention element concentration variations in cpx in relation to albite lamellae. While there is little doubt that albite formed by precipitation from the host cpx, it would be much more interesting to provide such data for cpx with amp lamellae. After all, the formation of amp lamellae is one of the issues in the contribution.

The reason for including the mineral chemical variation in clinopyroxene of sample DS2217 as Figure 5 is to highlight presumably hydrous phases of sub-micron width occurring in peripheral grain domains. These phases appear to be responsible for the absorption around $3622\,cm^{-1}$, which is absent in most spectra of the other samples (Figure 3a). However, we have added the requested element maps (Supplementary Figure 2). They show compositional halos of Al depletion and Ca enrichment around oriented inclusions of quartz + amphibole, which can be seen as an independent argument for an in-situ origin of these inclusions and to which we refer in subsection 5.4.

(16) lines 183-190: Here, the authors evaluate the significance of the ambiguous band at 3618-3633 cm-1 in cpx, which they ascribe to inclusions of sheet silicates (similar as in a paper by Koch-Müller et al. 2004). It would be interesting to know, which mineral in particular could be responsible. It was shown that phengite could be behind such absorption bands. Is there any phengite in the samples?

The answer to the referee's question is already included in the manuscript (section 2). Although sample 1066b from Fjørtoftvika, reported by Terry et al. (2000), contains phengite and comes from the same outcrop as sample DS1438, we could not detect phengite in any of our samples, but secondary biotite (in the matrix).

(17) line 209: The concentrations of rim and core (4–15 µgg−1 and 5– 18 µgg−1) are identical within the limits of uncertainty.

The referee is correct in that the ranges of the measured hydroxyl contents in the grain cores and grain rims are similar in the entire data set of orthopyroxene and overlap in terms of uncertainty. However, the referee seems to have overlooked the systematic nature of the data referred to in the following sentence. Individual analyses of cores and rims show systematically lower hydroxyl contents in rims compared to cores (Figure 6b). The concentration ratios between rims and cores based on average values are all below unity (Figure 7). We have added Supplementary Figure 3, which shows a single orthopyroxene crystal with analytical positions and quantified hydroxyl contents, to illustrate that the core and rim values differ outside the analytical uncertainty for a single crystal orientation.

(18) lines 212-213: Are all seven bands present in all samples? From Fig. 3, this does not seem to be the case. If not, is there a correlation with mineral composition or the paragenesis?

The answer to the first question is given in Table 1, which shows that the number of bands in each sample is not always 7 and varies between 4 and 7 depending on the sample and the analysed grain domain (area). As also shown in Table 1, the presence of certain bands in garnet does not depend on the presence of orthopyroxene in the mineral assemblage, subdividing the data set and reflecting two distinct mineral chemical trends of garnet (see Figure 6a, b in Spengler et al., 2023). Therefore, we see no evidence of a clear correlation between the bands and the mineral assemblage or the gar-net chemistry. However, when looking at the total number of missing bands in the entire data set, the orthopyroxene-bearing eclogite has fewer than the orthopyroxene-free eclogite. Whether this is coincidental or significant seems to be beyond the scope of this study.

(19) lines 251-254: The authors state that orthopyroxene rims show lower H2O contents than cores and thus experienced late hydrogen loss.
However, this is not quite supported by the results in Table 1. The differences are not significant be-cause they do not exceed the limits of uncertainty (at least 20 %). For cpx, there is a significant differ-ence between cor and rim composition in 5 samples. In two samples, rim and core compositions are identical. But this is not discussed.

We appreciate the critical view of the referee and would like to refer to our response to comment (17). The new Supplementary Figures 3 and 4 demonstrate that the difference in hydroxyl content between the core and the rim of single crystals lies outside the analytical uncertainty. Even if the average val-ues of the core and rim measurements of different crystals (with different crystal orientations) show a higher overall uncertainty, the relative difference between core and rim concentrations in the average data remains for all orthopyroxenes and many clinopyroxenes (Figure 7). In Figure 9, using the two polynomial approximations for clinopyroxene data, we now additionally show that this relative differ-ence increases with decreasing pressure. It shows that the systematic nature of the data is greater than the statistical effects resulting from different crystal orientations. If, on the other hand, the referee's concern were significant, these observations would be by chance.

(20) lines 273-274: The term 'inherited' is misleading. Are contents related to peak metamorphism or inherited from a prograde stage? Clarify.
Obviously, the use of the term "inherited" leads to confusion with a presumed prograde history (for which we provide no evidence). For the sake of clarity, we have reworded the title and parts of the subsection.

(21) lines 275-278: What caused the exceptionally high H2O in garnet of sample 1438?
Our answer is given in the same lines: "... the high structural hydroxyl content in its garnet is most likely related to the more hydrous conditions for the whole-rock during peak metamorphism." In principle, two possibilities are conceivable: Either the composition of the whole-rock (outcrop) of sample DS1438 differs in that it was inherently more hydroxyl-rich during UHP metamorphism, or the whole-rock was exposed to local fluids during UHP metamorphism, unlike the other samples. However, the subject of this study is not the cause of the exceptionally high hydroxyl content in the garnet of sample DS1438. More important, this sample shares oriented mineral inclusions in clinopyroxene with the other samples, suggesting that the observed mineral microstructure appears to be independent of differences in hydrous conditions (either "inherited" from whole-rock chemistry during crystallisation at the peak of metamorphism or caused by local fluids during crystallisation at the peak of metamorphism). The main conclusions are drawn from the other samples. The data does not seem to be able to provide a deeper answer, although it would be interesting to explore this, we fully agree. We have added explanatory sentences to subsection 5.2 to make this clearer.

(22) Fig. 8: Informative plot. Misprint in caption (exlogite).
We have expanded the information in the diagram and corrected the printing error in the caption.

(23) line 282: Simply say what you mean by 'independent mineral-chemical evidence for equilibration at UHP metamorphic conditions'. Opx-barometry, coesite?
The sentence is now reformulated using the term orthopyroxene barometry.

(24) lines 283-284: What are the properties of 'All other samples'. Are they non-UHP, or indication of non-equilibrium?
The referee's comment draws our attention to a point that should be better explained. Following the logic, "All other samples" contain either retrogressed orthopyroxene (i.e. there is evidence of retrogression) or no orthopyroxene (they differ in overall rock chemistry from the orthopyroxene-bearing eclogites). The point is that, prior to exsolution of quartz, orthopyroxene-bearing eclogite appears to have been more "wet" compared to orthopyroxene-free eclogite (with the exception of DS1438). Thus, the former could, during initial decompression from the stability field of Ca-Eskola & coesite to that of quartz, exsolve amphibole (together with quartz) while the latter did not (quartz only). Subsequent decompression increased the hydroxyl content of clinopyroxene in those samples that accumulated retrogression. We have added additional sentences to this part of the manuscript for clarity.

(25) lines 284-285: Here you conclude that hydroxyl was modified by the post-peak metamorphic evolution. The evidence for this - cpx in the zo-eclogite has not the highest H2O content (in contrast to garnet) - is not convincing. This may be the case, but without giving the mineral composition this is not justified. The high H2O content in garnet could just be due to high Ca.
In our response to comment (8), we explain that the Ca content in the garnet does not provide evidence that it can be the cause of the high hydroxyl content in the garnet of sample DS1438. The difference in hydroxyl content between clinopyroxene cores and rims (Figure 7, Supplementary Figure 3) increases with decreasing pressure (Figure 9) and may now provide more convincing evidence that the post-peak metamorphic evolution has altered the hydroxyl content in clinopyroxene.

(26) line 288: … opx, which shares a mineral paragenesis with garnet. What do you mean? Rephrase.
Unfortunately, we did not find any difficulties in understanding this sentence. We have therefore kept it.

(27) lines 302-308'However, the samples with the highest hydroxyl content in clinopyroxene of orthopyroxene-free eclogite (DS2217) and orthopyroxene-bearing eclogite (DS2216) are exposed only a few hundred metres apart (Figure 1). This suggests that the structural hydroxyl variation in orthopyroxene-free eclogite is in part also related to different degrees of retrogression. It follows from the spatial proximity of chemically different samples with highest structural hydroxyl content in clinopyroxene, their lowest metamorphic pressure estimates (Figure 9) and independent textural and mineralogical evidence for strong retrogression of these samples (Spengler et al., 2023)' etcRephrase.
The quoted sentence is indeed long and has now been reworded.

(28) line 316: What is precursor cpx? You mean the host cpx that previously contained more H2O?
Yes. The clinopyroxene that now hosts the oriented amphibole inclusions used to be without them, as amphibole is not stable under UHP conditions. The same logic applies to the oriented quartz inclusions, as quartz is also not stable at UHP and the orthopyroxene-bearing eclogites do not contain grains of free silica (i.e. $SiO_2$ was not part of the metamorphic mineral assemblage, making "overgrowth" of $SiO_2$ during clinopyroxene growth unlikely). Thus, quartz + amphibole in clinopyroxene are of secondary origin, for which two main scenarios can be considered: either the source of the chemical components that formed the oriented inclusions is external or internal. In the latter case, the (solid solution) precursor clinopyroxene is expected to be more hydrous than the current (exsolved) host clinopyroxene.

(29) lines 317-319: This reference to opx in order to exclude an external fluid rather farfetched. It is much more logical that fluid from an external source, first of all, reacts with the rim of cpx to form matrix pargasite by consumption of cpx rims. Such pargasite contains inclusions of vermicular quartz as seen in numerous examples of UHP eclogite. If such features are not observed, an external fluid is highly unlikely.
The referee puts forward an interesting alternative hypothesis, which is challenged by the following remarks:
(i) Amphibole, which occurs as lamellae together with quartz in clinopyroxene grain cores, has a lower K content than amphibole grains in the matrix (see reply to comment (6)).
(ii) Oriented inclusions of quartz+amphibole in clinopyroxene were never observed exclusively at the rims of the clinopyroxene grains, as would be expected if they originated from an external fluid source. Instead, they occur either homogeneously in whole clinopyroxene crystals (e.g. sample DS1438 in Figure 4c in Spengler et al., 2023) or with variable density in the grain cores, while they disappear towards the grain rims (e.g. sample M65 in Figure 2c, e in Spengler et al., 2023).
(iii) All oriented quartz inclusions in clinopyroxene that occur together with amphibole have a crystallographic orientation relationship with the clinopyroxene host (recognisable by a common angle of extinction under crossed-polarised light).
(iv) The hydroxyl loss in orthopyroxene rims (Figure 7) seems to be inconsistent with an infiltration by an external fluid.
(v) It is highly unlikely that a penetrating fluid that partially alters the core of a clinopyroxene crystal would leave the core of a neighbouring orthopyroxene crystal untouched, so that the latter retains a very low alumina content, indicating metamorphic conditions in the stability field of diamond, i.e. where amphibole is not stable.

(30) lines 322-323: This argument is good.
This argument complements the remarks against the referee's hypothesis proposed in comment (29).

(31) line 333: The volume of such lamellae is often overestimated. One per cent presumably is an upper limit. A rough estimation is easily done in BSE images.
We have modified the sentence and refer to a BSE image in the supplement (Supplementary Figure 1b). In addition, the interested reader can obtain chemically integrated analyses of the microstructure from the earlier study (Spengler et al., 2023).

(32) line 342: '…samples thought to have formed by metasomatism…' Which samples? Reference?
The referee's question has been answered by repeating the reference from line 337.

(33) lines 353-355: 'The source of the hydrous fluid is… likely related to partial melting of eclogite…' Unlikely. Partial melting draws all the fluid out of the rock because H2O strongly partitions into the melt. As a result, H2O will be depleted in the solid restite and not enriched.
The referee is right, and we fully agree that partial melting is likely to result in partitioning of $H_2O$ into the melt. In response, we replaced the passage on melting with a sentence suggesting that the infiltration of fluids can explain the crystallisation of hydrous phases such as the matrix biotite, which is a source of rheological weakening and thus strain partitioning.

**References in the reply**

Anderson, E. D. and Moecher, D. P. (2007). Omphacite breakdown reactions and relation to eclogite exhumation rates. *Contributions to Mineralogy and Petrology*, 154(3):253–277.

Bromiley, G. D. and Keppler, H. (2004). An experimental investigation of hydroxyl solubility in jadeite and na-rich clinopyroxenes. *Contributions to Mineralogy and Petrology*, 147(2):189–200.

Gose, J. and Schmädicke, E. (2022). H2o in omphacite of quartz and coesite eclogite from erzgebirge and fichtelgebirge, germany. *Journal of Metamorphic Geology*, 40(4):665–686.

Katayama, I. and Nakashima, S. (2003). Hydroxyl in clinopyroxene from the deep subducted crust: evidence for $H_2O$ transport into the mantle. *American Minerlogist*, 88(1):229–234.

Koch-Müller, M., Matsyuk, S. S., and Wirth, R. (2004). Hydroxyl in omphacites and omphacitic clinopyroxenes of upper mantle to lower crustal origin beneath the siberian platform. *American Mineralogist*, 89(7):921–931.

Konzett, J., Libowitzky, E., Hejny, C., Miller, C., and Zanetti, A. (2008). Oriented quartz+calcic amphibole inclusions in omphacite from the saualpe and pohorje mountain eclogites, eastern alps—an assessment of possible formation mechanisms based on ir- and mineral chemical data and water storage in eastern alpine eclogites. *Lithos*, 106(3–4):336–350.

Spengler, D., Włodek, A., Zhong, X., Loges, A., and Cuthbert, S. J. (2023). Retrogression of ultrahigh-pressure eclogite, western gneiss region, norway. *European Journal of Mineralogy*, 35(6):1125–1147.

Terry, M. P., Robinson, P., and Krogh Ravna, E. J. (2000). Kyanite eclogite thermobarometry and evidence for thrusting of UHP over HP metamorphic rocks, Nordøyane, Western Gneiss Region, Norway. *American Mineralogist*, 85:1637–1650.

---

## Author Response (AR2)

Dr. Dirk Spengler
Faculty of Geology, Geophysics and Environmental Protection
AGH University of Science and Technology
al. Mickiewicza 30
30-059 Kraków
POLAND

Prof. Dr. Andrea Di Muro
Laboratoire de Géologie de Lyon: Terre, Planètes, Environnement
Université Claude Bernard Lyon 1
bâtiment Géode (Campus Doua)
69622 Villeurbanne Cedex
FRANCE

Berlin, 10th January 2025

**COVER LETTER: egusphere-2024-2734 – minor corrections**

Dear Professor Di Muro,

Thank you for accepting our manuscript entitled "Hydroxyl in eclogitic garnet, orthopyroxene and oriented inclusion-bearing clinopyroxene, W Norway" for publication in *Solid Earth*.

As you requested, we have checked the use of the molecular weight for water in equation 1. In our opinion, this equation is correct. It refers to the rounded molecular weight of $H_2O$, which is multiplied by 0.1 to obtain the result in weight percent.

We have corrected the typo you found and also added the letter "S'' to the numbering of the tables and figures in the supplementary information to match the style of the journal.

Best regards,

Dr. Dirk Spengler